# Whose name is it anyway? Varying patterns of possessive usage in eponymous neurodegenerative diseases

Michael R. MacAskill[1,2] and Tim J. Anderson[1,2,3]

[1] New Zealand Brain Research Institute, Christchurch, New Zealand
[2] Department of Medicine, University of Otago, Christchurch, New Zealand
[3] Department of Neurology, Christchurch Hospital, Christchurch, New Zealand

## ABSTRACT

There has been long-standing debate over whether use of the possessive form of the names of eponymous neurological disorders should be abandoned. Which view has actually predominated in practice? We empirically assessed current and historical usage in the scientific literature. The PubMed database was queried for the percentage of titles published each year from 1960–2012 which contained the possessive form of Parkinson's (PD), Alzheimer's (AD), Huntington's (HD), Wilson's (WD), and Gaucher's (GD) diseases (e.g. Huntington's disease or chorea vs Huntington disease or chorea). Down syndrome (DS), well known for its changes in terminology, was used as a reference. The possessive form was nearly universal in all conditions from 1960 until the early 1970s. In both DS and GD it then declined at an approximately constant rate of 2 percentage points per year to drop below 15%. The possessive forms of both PD and AD began to decline at the same time but stabilised and have since remained above 80%, with a similar but more volatile pattern in HD. WD, meanwhile, is intermediate between the DS/GD and PD/AD/HD patterns, with a slower decline to its current value of approximately 60%. Declining possessive form usage in GD and DS papers has been remarkably uniform over time and has nearly reached completion. PD and AD appear stable in remaining predominantly possessive. The larger volume of papers published in those fields and their possibly greater public recognition and involvement may make that unlikely to change in the short-term. In a secondary analysis restricted to PD, we found that practices have switched dramatically several times in each of three US-published general neurology journals. Meanwhile, in two UK-published journals, and in the specialist title "Movement Disorders", the possessive form has been maintained consistently. The use of eponyms in neurology shows systematic variation across time, disorders, and journals.

## INTRODUCTION

Neurology has an extensive tradition of naming disorders after their putative discoverers, although this practice is now actively discouraged in favour of descriptive or explanatory

Corresponding author
Michael R. MacAskill,
michael.macaskill@nzbri.org

names. A substantial legacy of historical eponyms remains, however, and it has even been noted that their opacity can be an advantage if initial explanatory or descriptive names are subsequently shown to be incorrect (*McKusick, 1998*). For example, Dr. James Parkinson was not so immodest as to name a disorder after himself, and originally termed it *paralysis agitans* on the basis of the symptoms he observed. It was, however, re-named in honour of Parkinson following Charcot's demonstration that weakness (paralysis) is not actually a feature of the disorder and that tremor (agitans) is not universally present (*Goetz, 1986*). Charcot was generally complimentary towards Parkinson's work, but thought the disorder had probably been described earlier by a French physician (*Goetz, 1986*), while others believe it was described in India long before (*Ovallath & Deepa, 2013*). This is not an isolated instance: according to (the ironically-named) Stigler's Law of Eponymy, no scientific term is named after its original discoverer (*Stigler, 1980*).

Despite criticisms, medical eponyms remain in common use, although debate has also continued about how they should be employed. There have been strident arguments since the mid 1970s advocating either the abolition or the retention of their possessive form. For example, "Parkinson disease" has been proposed as a more grammatically accurate term than "Parkinson's disease", because Parkinson neither had, nor owned, the disease associated with his name (*Haines & Olry, 2003*; *Smith, 1975*). In general, those in favour of retaining the apostrophe simply appeal to the value and comfort of traditional practice, although some have also made sophisticated grammar-based counter-arguments (*Dirckx, 2001*).

Formal style guidelines, such as those from the World Health Organisation (*World Health Organization, 2004*) and the American Medical Association (*Iverson, Christiansen & Flanagin, 2007*), have long advocated the elimination of the possessive form. It remains to be shown how influential those recommendations have been in neurology. How have naming practices altered over time? Is usage continuing to change or has it reached a steady state? Do conventions vary across disorders or journals? To assess this, we conducted a quantitative analysis of the titles of scientific papers published between 1960 and 2012. We focussed selectively upon our own interests in neurodegenerative conditions, particularly those with a movement component (although the scripts we used to gather and analyse the data are available as Supplemental Information for anyone who might wish to apply them in other domains).

This evidence was not gathered to advocate for either viewpoint. Having encountered differing sub-editing practices across journals, we simply wished to empirically ascertain their relative predominance, currently and historically. By way of full disclosure, however, we ourselves lean towards the traditionalist view, while believing that if anyone can be said to "own" a disease, it is those patients who have it. Decisions on disease nomenclature should ideally not be made in isolation within academic neurology, but only with full involvement of patients' support, advocacy, and fund-raising organisations. Changes in naming practice apply not only to a pathological process but can also be linked closely to the self-identity of patients.

## MATERIALS & METHODS

The neurodegenerative diseases we selected were Parkinson's (PD), Alzheimer's (AD), Huntington's (HD), Wilson's (WD), and Gaucher's (GD). The congenital Down syndrome (DS), well known for a history of lability in its nomenclature, was chosen as a reference for comparison. We adapted a publicly available script (*Magnusson, 2012*), running within the open source R statistical environment (*R Development Core Team, 2013*), to make automated queries of the PubMed database. We extracted a count of the titles of published articles which contained the possessive or non-possessive form of each name (for example, "Parkinson's disease" vs "Parkinson disease"), for each year from 1960–2012. The outcome measure was the percentage of papers which used the possessive form, calculated for each disorder within each year. Non-eponymous alternatives (such as hepatolenticular degeneration, trisomy 21) were not considered. Searches for HD and AD included "chorea" and "dementia" respectively as alternatives to "disease". An example pair of queries is "Huntington's disease[TITLE] OR Huntington's chorea[TITLE]" vs "Huntington disease[TITLE] OR Huntington chorea[TITLE]". To avoid the double-counting of papers which were initially published electronically but subsequently appeared in print in the following calendar year, the date was specified using the Print Dates Only Tag [PPDAT] rather than Publication Date [DP] (*Magnusson, 2012*). Queries were intentionally limited to a rate of less than 3 per second to comply with current PubMed guidelines for automated access. The dataset, obtained on 25 January 2013, took approximately 11 min to collect. The R script used to obtain the data, and the resulting dataset itself, are available in the Supplemental Information.

We then investigated how patterns vary across a number of prominent primary research journals. Because the number of papers published annually on a given disorder in a particular journal is usually small, we restricted the analysis to the two most frequently published disorders (Parkinson's and Alzheimer's) to reduce instability in the proportion measure. Their results were strikingly similar and for clarity we present only the Parkinson's findings. We restricted the data collection to each of selected general neurology journals published either in the United Kingdom *(Brain; Journal of Neurology, Neurosurgery & Psychiatry)* or the United States *(Neurology; Archives of Neurology; Annals of Neurology)*, because different national practices have been noted previously (*Jana, Barik & Arora, 2009*). We also assessed *Movement Disorders*, the official journal of the international Movement Disorder Society. As a specialist journal, it publishes more Parkinson's-titled papers than these general journals combined (in 2012, 176 papers vs 127).

No ethical approval was either sought or required for this study as it involved only publicly-available data.

## RESULTS

In the first analysis (across all journals), the total number of papers identified was 82 249 (AD: 33 768; PD: 26 754; DS: 10 569; HD: 5 750; WD: 2 935; GD: 2 473). The proportion of paper titles containing the possessive form of each disorder is plotted as a function of time in Fig. 1. The data are summarised using second-order LOESS local regression with a

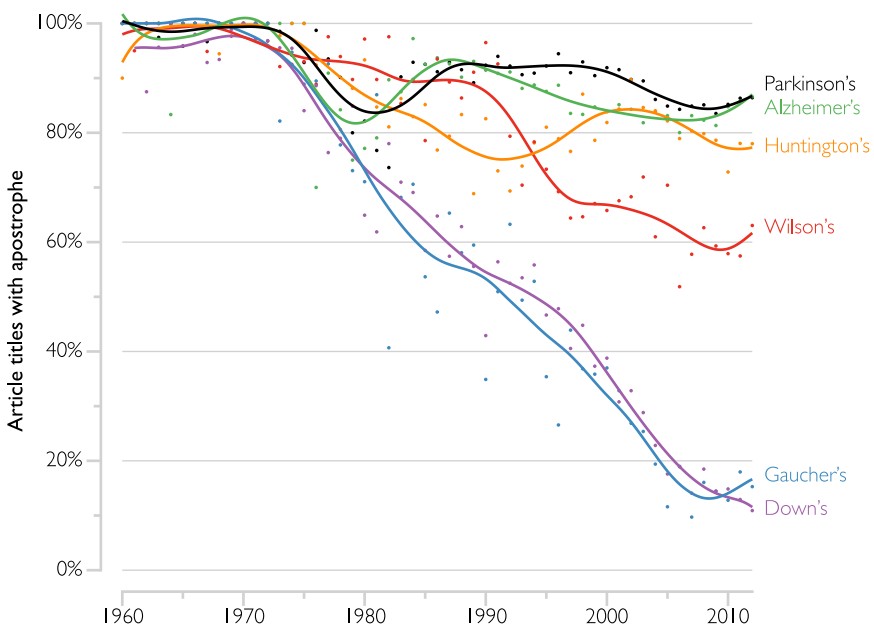

**Figure 1 Changing proportion of article titles using the possessive form of selected eponymous neurodegenerative diseases.** LOESS localised regression curves showing the time-varying proportion of paper titles containing the possessive forms of five neurodegenerative disorders, compared to Down syndrome. The possessive form remains dominant in Parkinson's, Alzheimer's, and Huntington's, whereas it is becoming extinct in Gaucher disease. Wilson's disease shows an intermediate pattern of slow decline.

smoothing width of 0.2, as implemented in DataGraph (*MacAskill, 2012*, Visual Data Tools Inc.).

The possessive form was nearly universal in all of the disorders from 1960 until the early 1970s. From then, the possessive forms of both DS and GD began to decline at a nearly constant rate of approximately 2 percentage points per year to reach their current prevalence (≤15%). Use of the possessive forms of both PD and AD began to decline at a similar time but ceased doing so from approximately 1980 and it has remained above 80% ever since. The pattern in HD, while more volatile, is similar to that of PD and AD. With a slow decline to its current value of approximately 60%, WD is intermediate between the DS/GD and PD/AD/HD patterns.

The results of the second, journal-specific, analysis of Parkinson's-titled papers are shown in Fig. 2. The UK-based general journals *Brain* and *Journal of Neurology, Neurosurgery & Psychiatry* show consistent and nearly universal use of the possessive form over the entire period 1960–2012. *Movement Disorders* shows a similar pattern, from its commencement of publication in 1986. It is not clear whether this reflects consistent enforcement of editorial practice by these journals or the preferences of authors. The US-based journals show much greater lability, each rapidly alternating several times from high to low use. The rapidity of these changes implies that the cause was likely due to editorial practice, as author submissions are unlikely to alter so swiftly and universally. The changes are also lagged in time, with *Archives of Neurology* switching first on each occasion, followed by *Neurology* and then *Annals of Neurology*. This again indicates that the changes

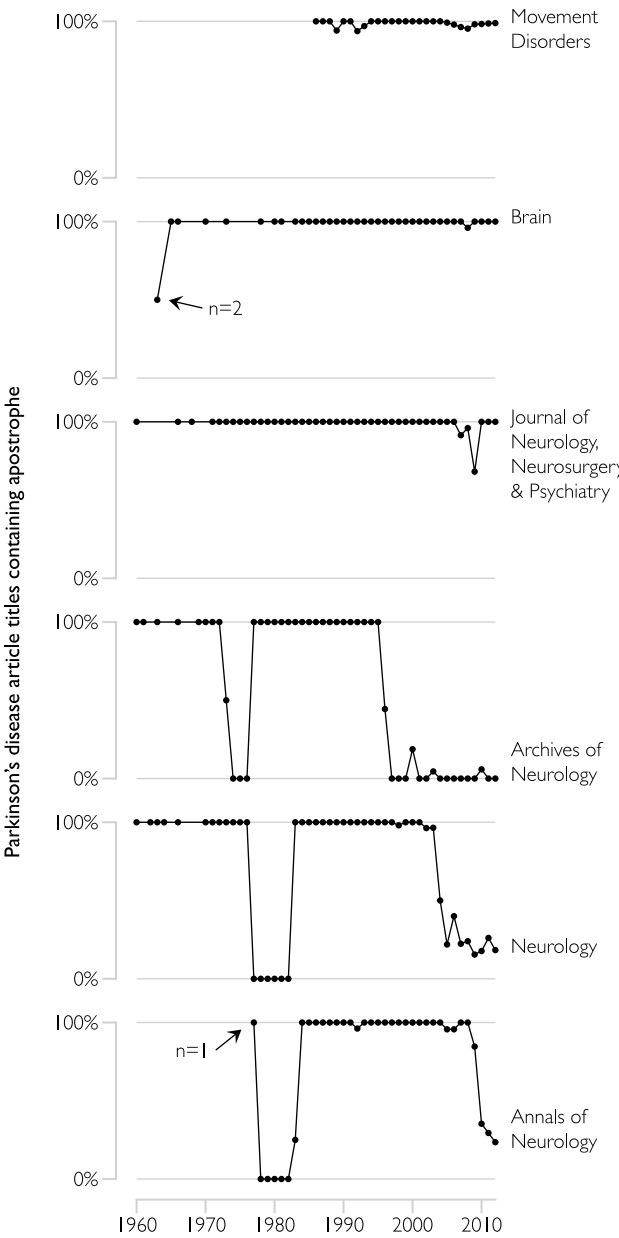

**Figure 2  Proportion of article titles containing the possessive form of "Parkinson's disease" in prominent neurological journals.** The analysis shown in Fig. 1 for Parkinson's disease was repeated for each of six selected journals. In all journals, the pattern for Alzheimer's disease was almost identical to that shown here for Parkinson's. The possessive form has predominated consistently in the specialist journal *Movement Disorders* and the UK-based generalist *Brain* and *Journal of Neurology, Neurosurgery, and Psychiatry*. The other, US-based journals, meanwhile, have oscillated several times between high and low usage, indicating discrete changes in editorial policy. *Movement Disorders* and *Annals of Neurology* commenced publishing in 1986 and 1977 respectively. Gaps between data points in the early period of the time series for the other journals correspond to years when no Parkinson's papers were published.

were unlikely to be driven by changes in author preferences, as these should have affected each of the journals simultaneously.

We inspected the instructions to authors posted on each journal's website. None explicitly addressed eponym usage. *Archives of Neurology* advises authors to prepare manuscripts according to the AMA Manual of Style (which advocates the non-possessive form) or the ICMJE (International Committee of Medical Journal Editors) 'Uniform Requirements for Manuscripts Submitted to Biomedical Journals', which is silent on the issue (*Jana, Barik & Arora, 2009*). *Movement Disorders, Brain,* and *Neurology* each also point to the ICMJE, even if only for reference formatting rather than as a general style guide.

## DISCUSSION

Recommendations to abandon the use of possessive eponyms were made from 1974 by the US National Institutes of Health (*Lowry, 1974*; *Smith, 1975*). It is perhaps not coincidental that our data show that the initial decline in possessive usage commenced at about that time, and that shortly afterwards it disappeared completely (if temporarily) in the three US journals we examined. Reduction in the possessive form of GD has been remarkably uniform since then, has nearly reached completion, and has closely paralleled that of DS. By contrast, PD and AD rebounded from the initial decline, and now appear to be remaining relatively stable in their predominantly possessive forms. That may be unlikely to change in the short term, given the inertia due to the much larger volume of work published in those disorders and their likely greater public recognition and involvement. Judged solely by numbers of publications, the other disorders presumably have smaller scientific communities and are therefore perhaps more amenable to systematic change (in the case of GD and DS) or susceptible to more volatility (HD and WD).

Mathematical models have recently been proposed which examine how changes in a social consensus can occur, given varying-sized initial minorities of influential proselytisers. It has been claimed that the time required for the entire population to change its view is dramatically decreased once the proselytising proportion exceeds a critical value of 10% (*Xie et al., 2011*). Our data might argue against the generality of that claim, perhaps indicating that larger communities are more resistant to change. It must be borne in mind though, that the proportion of published papers is certainly not a perfect proxy for the proportion of researchers who adhere to either point of view. That is, even after non-possessive usage exceeded 10% in AD and PD papers, that may have been due only to involuntary compliance with individual editorial guidelines rather than to proselytizing zeal on the part of researchers themselves. Hence the practice has not continued to grow. By contrast, the change may have been internalised by researchers in the GD and DS spheres. As noted, none of the journals we surveyed lists an explicit guideline on eponym usage, and in our experience, accepted manuscripts are simply altered without notice at the sub-editing stage (for example, *Dalrymple-Alford et al., 2010*).

With reference to the rapidly declining use of the Hallervorden-Spatz eponym due to its association with Nazi war crimes, *Shevell (2012)* noted that "...there is no central

body regulating the use of eponyms. Any changes in designation or use thus must reflect a naturally occurring, emerging, and broadly based consensus...a reflection of a decision by a neurologic 'court' of opinion". Based on our comparisons across a number of disorders and journals, it would appear there is more than one such court in session, and that the judgments of one do not necessarily act as binding precedent upon another.

### Funding
No funding supported this study other than the salaries provided by the authors' employers.

### Competing Interests
The authors have no conflicts of interest with regards to the reported findings or conclusions.

### Author Contributions
- Michael R. MacAskill conceived and designed the experiments, analyzed the data, wrote the paper.
- Tim J. Anderson interpreted the data, reviewed and critiqued the manuscript.

### Supplemental Information
Supplemental information for this article can be found online at http://dx.doi.org/10.7717/peerj.67.

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
