# Peer review of "Whose name is it anyway? Varying patterns of possessive usage in eponymous neurodegenerative diseases"

_PeerJ, doi:10.7717/peerj.67_

## Round 0.1 · original submission · Minor Revisions

As a clinician, I find that making a diagnosis of paralysis agitans is fairly straightforward but the eponymic designation (Parkinson's or Parkinson disease) is not. The authors nicely demonstrate that there is no prevailing consensus for the 6 eponymous diseases which they have studied. Nor is there a compelling philological rule for naming scientific discoveries. The surname Parkinson has been associated with scientific findings other than the "Shaking Palsy" such as (Dwight) Parkinson's Triangle (in the cavernous sinus) and (C. Northcote) Parkinson's Law ("work expands so as to fill the time available for its completion'). Usage of the possessive eponym seems to prevail in these realms of anatomy and economics but most likely reflects the caprices of usage rather than any principles of philology.
This succinct paper nicely shows that eponymous nosology is a social (and not rigorously scientific) construct. And that's helpful information when I'm confronted with the Parkinson/Parkinson's option.

Please address the reviewer's 5 comments/queries. Also consider using the graph with raw data points for Figure 1.

·

Basic reporting

The authors could improve the introduction by adding a sentence such as "Some prominent journals have a stated policy on this question; e.g. Neurology, Brain, ... require the name without the 's." or whatever is actually the case. They may also consider adding a reference to a book of eponyms, or to the ironically eponymous Stigler's law: "No scientific discovery is named after its original discoverer". These comments are meant as suggestions, not requirements.

Experimental design

No comment.

Validity of the findings

Technically it could be observed that the authors do not formally test for differences between diseases in the time curves plotted in the Figure. Nevertheless, for this article, the primary conclusions are limited, and the sample sizes are large, and the results are rather obvious, so I believe the conclusions are defensible without such a statistical test.

Comments for the author

This is a delightful article, short and to the point, with data to shed light on a topic that, if not of crucial significance, is at least of great interest to clinicians and researchers in neurology and psychiatry.

Reviewer 2 ·

Basic reporting

ok

Experimental design

ok

Validity of the findings

ok

Comments for the author

This is an excellent paper on a quirky topic. I would definitely recommend publication with only minor revisions and no need for me to re-review.

Items for the authors to consider:



1. They may consider also looking at article abstracts and article key words in addition to titles.

2. Did they restrict this to English language only?

3. Consider stratifying by type of paper: review, original article, case report.

4. It may be interesting to report the results for the 5 - 10 highest impact journals, to see whether they are traditionalists or not.

5. "fewer than 3 per second" or "less than 3 per second"?

---

## Round 0.2 · accepted · Accept

The minor revisions suggested by one referee have been nicely addressed.
My sincere hope is that scientific editors will take note of this elegant little study and avoid imposition of stylistic "one size fits all" rules for eponym usage.